# Physicochemical and Antioxidant Properties of Nanoliposomes Loaded with Rosemary Oleoresin and Their Oxidative Stability Application in Dried Oysters

**DOI:** 10.3390/bioengineering9120818

**Published:** 2022-12-19

**Authors:** Xiaoyu Cheng, Mingwu Zang, Shouwei Wang, Xin Zhao, Guozhen Zhai, Le Wang, Xiang Li, Yan Zhao, Yijing Yue

**Affiliations:** 1Beijing Key Laboratory of Meat Processing Technology, China Meat Research Center, Beijing Academy of Food Sciences, Beijing 100068, China; 2College of Food Science and Nutrition Engineering, China Agricultural University, Beijing 100083, China

**Keywords:** rosemary oleoresin, nanoliposomes, antioxidant, dried oysters

## Abstract

Lipid and protein oxidation is a main problem related to the preservation of dried aquatic products. Rosemary oleoresin is widely used as an antioxidant, but its application is limited due to its instability and easy degradation. Nanoliposome encapsulation is a promising and rapidly emerging technology in which antioxidants are incorporated into the liposomes to provide the food high quality, safety and long shelf life. The objectives of this study were to prepare nanoliposome coatings of rosemary oleoresin to enhance the antioxidant stability, and to evaluate their potential application in inhibiting protein and lipid oxidation in dried oysters during storage. The nanoliposomes encapsulating rosemary oleoresin were applied with a thin-film evaporation method, and the optimal amount of encapsulated rosemary oleoresin was chosen based on changes in the dynamic light scattering, Zeta potential, and encapsulation efficiency of the nanoliposomes. The Fourier transform-infrared spectroscopy of rosemary oleoresin nanoliposomes showed no new characteristic peaks formed after rosemary oleoresin encapsulation, and the particle size of rosemary oleoresin nanoliposomes was 100–200 nm in transmission electron microscopy. The differential scanning calorimetry indicated that the nanoliposomes coated with rosemary oleoresin had better thermal stability. Rosemary oleoresin nanoliposomes presented good antioxidant stability, and still maintained 48% DPPH radical-scavenging activity and 45% ABTS radical-scavenging activity after 28 d of storage, which was 3.7 times and 2.8 times higher than that of empty nanoliposomes, respectively. Compared with the control, the dried oysters coated with rosemary oleoresin nanoliposomes showed significantly lower values of carbonyl, sulfhydryl content, thiobarbituric acid reactive substances, Peroxide value, and 4-Hydroxynonenal contents during 28 d of storage. The results provide a theoretical basis for developing an efficient and long-term antioxidant approach.

## 1. Introduction

Oysters are one of the most economically valuable aquaculture products, with good flavor and high nutritive values, consisting of high-quality protein, polyunsaturated fatty acids (PUFA), functional substances, and several other micronutrients [1]. Oyster production accounts for about one-third of global marine bivalve production, constituting an important marine resource; they are the most cultivated bivalves [2]. Dried oysters, one of the most common seafood products, are processed by adopting an important traditional processing method, which extends the storage period of oysters and produces a characteristic flavor [3]. However, lipid and protein oxidation is a main problem related to dried oyster preservation, which results not only in the deterioration of food quality and economic losses, but also in the production of harmful compounds with mutagenic, cytotoxic, and oxidative effects on body tissues; thereby promoting inflammation, atherosclerosis, cancer, and aging processes [4,5]. Notably, 4-hydroxy-2-nonenal (HNE), which is mainly produced by peroxidation of PUFA, is a harmful aldehyde that accumulates in the body, and is particularly difficult to metabolize and excrete [6]. In order to minimize the risk of lipid and protein oxidation, antioxidants are widely used for food processing and storage [7]. Synthetic antioxidants are widely used in the food industry to prolong the shelf life and have the oxidation-inhibiting effects of food; however, their use is currently decreasing due to their suspected genotoxicity. Considering human health and environmental preservation, the demand for natural antioxidants is increased to delay lipid and protein oxidation, prolong shelf life, and promote sustainability [8].

In recent years, plant extracts have shown an obvious effect on antioxidant properties, and the demand for plant components as additives is increasing [9,10]. Rosemary is a widely used natural spice, and the antioxidant activity of oleoresin extracted from dried rosemary leaves in food has been proven [11,12,13,14]. The antioxidant properties of rosemary are due to its phenolics and terpenes, such as carnosic acid, carnosol, α-pinene, and others, which have high radical-scavenging activity [15]. Rosemary spice is a potential functional ingredient of meat products, like beef, chicken, and pork, due to its excellent antioxidant and antibacterial properties [16]. Rosemary spice has many forms, such as rosemary essential oil, rosemary extract, rosemary resin, and ground rosemary [17]. Oleoresins are viscous liquids extracted from spices, which contain not only essential oil, but also some non-volatile components, such as pigment, fatty oils, and phenolic antioxidants. Compared to essential oils, oleoresins are possibly able to provide sustainable antioxidant capacity due to their low volatility. Therefore, in this study, rosemary oleoresin (RO) was used in the nanoliposome encapsulation. Rosemary oleoresins, as an active ingredient, are viscous liquids extracted from spices, which have lower volatility and longer anti-oxidation times than essential oils and extracts [12]. However, plant oleoresins are sensitive to temperature, air, light, and relative humidity, resulting in chemical instability and easy degradation. Some bioactive ingredients of plant oleoresins are highly hydrophobic and poorly water-soluble, reducing bioavailability and restricting their application [18]. 

Liposome encapsulation can decrease the loss of bioactives using membrane bilayers. It forms to provide effective photoprotection and oxidation degradation protection. Nanoscale structures can further enhance the dispersion of bioactives in water and improve its water solubility. Appropriate encapsulation of bioactives with micrometric-size capsules can improve the controlled release properties, and in general, does not affect the activities [19,20,21,22]. To date, the characterization and activities of nanoliposome-coated essential oils, such as cinnamon essential oil, *Satureja khuzestanica* essential oil, artemisia annua oil, thyme (*Thymus zygis*) essential oil, as well as encapsulated clove and lavender essential oil and others, have been relatively more studied [23,24,25,26]. However, comparatively little research is available about the characteristics and activities of nanoliposome-encapsulated plant oleoresins. Additionally, the current research focused more on the chemical and physiological characteristics of nanoliposomes [27,28,29,30,31,32], rather than the application in food. Furthermore, no data exist on the application of the edible coating containing RO-loaded nanoliposomes in dried oysters. 

The study aimed to organize RO nanoliposomes and estimate their stability parameters. Meanwhile, RO nanoliposomes’ characterization, including dynamic light scattering (DLS), Fourier transform-infrared spectroscopy (FTIR), microstructure, physical stability, antioxidant stability, and antioxidant in dried oysters were also investigated. The study delivers an efficient and pragmatic approach to enhancing food quality based on natural antioxidant coatings.

## 2. Materials and Methods

### 2.1. Materials

The same batch of frozen oysters from South China Sea, with good commodity quality were obtained from Raoping Yujie Food Industry Co., Ltd. (Guanngdong, China), transported to the laboratory through a cold chain, stored at −20 °C, then transferred to 4 °C the night before the experiment for defrosting. RO (the content of carnosic acid was 15.36%, *w/w*) was obtained from Kalsec Co, Ltd. (Shanghai, China). Fish 4-HNE ELISA Kit was obtained from Beijing Kangpu Tongchuang Co, Ltd. (Beijing, China). 2,2’-azino-bis (3-ethylbenzothiazoline-6-sulfonic acid) (ABTS) and 2,2-Diphenyl-1-picrylhydrazyl (DPPH) were obtained from Macklin Regent Co, Ltd. (China). Trichloroacetic acid (TCA) was from Shanghai Aladdin Reagent Inc. (Shanghai, China). Sodium dihydrogen phosphate dihydrate, 2-thiobarbituric acid (TBA), absolute ethanol (analytical reagent), Tween-80, and sodium dihydrogen phosphate dodecahydrate (Sinopharm Chemical Regent Co, Ltd., Beijing, China) were used.

### 2.2. Preparation of RO Nanoliposomes and Encapsulation Efficiency

Nanoliposomes were loaded with RO by the thin-film evaporation technique [32] with slight modifications. First, lecithin and cholesterol (4:1, *w/w*), mixed with Tween-80 in an equal weight of lecithin, were dissolved in the appropriate amount of absolute ethanol, then varying amounts of RO (13, 26, 39, 52 mg in 100 mg empty nanoliposomes) were added to the solutions and stirred at ambient temperature (25 °C) for 30 min. By rotary evaporation at 35 °C, the solvent was removed, and a thin film was formed on the flask walls under vacuum. The resulting film was placed into a drying basin for 24 h to dry and mixed with phosphate-buffered saline (PBS, pH 7.4). The mixture was stirred until the aqueous phase became milky. To decrease the particle size, the suspensions were subjected to ultrasonication (1 min/1 s, on/off pulse) at 30% amplitude. Thereafter, the suspension was passed through a 0.22-µm pore-size filter. The samples were freeze-dried overnight. The best encapsulation efficiency of the sample was chosen to determine the following physicochemical and antioxidant indexes.

Carnosic acid is one of the main antioxidant ingredients in RO [33]. The content of carnosic acid was measured to evaluate the antioxidant capacity of RO. Different concentrations of carnosic acid were measured by HPLC (Waters 2998, America) referring to the detection method described by Sui et al. [34]. The total amount of carnosic acid in the sample was measured by adding absolute ethanol to obtain encapsulated and unencapsulated carnosic acid. Then, the sample was placed in 10-kDa dialysis tubing for 24 h at 4 °C, and the dialysis solution was changed every 6 h to obtain encapsulated carnosic acid. Encapsulation efficiency of the sample was given by Equation (1):(1)$$\mathrm{Encapsulation}\;\mathrm{efficency}=\frac{\mathrm{RO}\;\mathrm{encapsulated}}{\mathrm{Total}\;\mathrm{RO}}\times 100\%$$

### 2.3. DLS Measurement 

DLS measured Brownian motion corresponding with particle size. The mean particle size and PDI, and zeta potential of samples were determined by DLS using a Zeta sizer Nano instrument (Malvern, UK). Nanoliposomes samples were diluted 100-fold and slowly injected into the cell to reduce the effect of bubbles and multiple scattering.

### 2.4. FTIR Measurement

Samples were combined with KBr and pressed to form disks. FTIR measurements were obtained by using spectroscopy (Thermo Fisher Scientific, Sunnyvale, CA, USA) in transmission mode, and the scanning range from 500 to 4000 cm^−1^. Each sample was measured three times.

### 2.5. Transmission Electron Microscopy (TEM) Measurement

TEM (H-7650, Hitachi, Tokyo, Japan) was used to visualize the image of the nanoliposomes and RO nanoliposomes. Samples were put on a copper grid to air dry for 5 min following dilution with water. Then, the grid was negatively stained with phosphotungstic acid (3 min). The excess solution was removed using filter paper. The grid was observed following drying at room temperature.

### 2.6. Differential Scanning Calorimetry (DSC) Analysis

The thermal stability of nanoliposome and RO nanoliposome were measured by DSC Q2000 (TA instruments, New Castle, DE, USA). Approximately 7 mg nanoliposomes and RO nanoliposomes were weighted and analyzed using DSC (0 to 300 °C, 5 °C/min increased). The nitrogen flow was 50.0 mL/min. 

### 2.7. Temperatures and pH Stability

The physical stability of nanoliposome was investigated at varying temperature (4 or 25 °C) and pH values (pH 2–12), which were adjusted to the desired pH values ranging from 2 to 12 by adding different amounts of HCl or NaOH solutions to the original samples. The particle size was determined.

### 2.8. Antioxidant Activity of RO

The antioxidant properties of RO (0.5 mg/mL) and RO nanoliposomes (0.5 mg/mL) were characterized by DPPH and the ABTS radical-scavenging activity, and were detected in triplicate. The DPPH radical-scavenging activity of RO and RO nanoliposomes were measured using the method of Sarabandi et al. [35], with slight modifications. 2 mLnanoliposome samples were mixed with 2 mL of 0.1 mM DPPH solution prepared in ethanol. The mixture was then vortexed and kept for 30 min in a dark place. Then, the absorbance of the mixture was measured at 517 nm, and was calculated using the following Equation (2):

The ABTS radical-scavenging activity of RO and RO nanoliposomes were measured referring to the method of Shishir et al. [36], with slight modifications. First, 2 mL of 7 mmol ABTS solution was combined with the same volume of 2.45 mmol potassium persulfate. The solution mixture was incubated in the dark for 12–16 h at room temperature. Before the test, 1 mL of the solution mixture was diluted with ethanol to obtain an absorbance rate of 0.70 ± 0.02 at 734 nm. Then, 3800 µL test mixture solution was added in 200 µL of nanoliposomes, reacted in the dark for 30 min, and measured at 734 nm. Then, the ABTS of RO and RO nanoliposomes were calculated from the following Equation (3):(2)$$\mathrm{DPPH}(\%)=\left(1-\frac{A\;\mathrm{amount}\;\mathrm{of}\;\mathrm{control}-A\;\mathrm{amount}\;\mathrm{of}\;\mathrm{sample}}{A\;\mathrm{amount}\;\mathrm{of}\;\mathrm{control}}\right)\times 100$$
(3)$$\mathrm{ABTS}(\%)=\left(1-\frac{A\;\mathrm{amount}\;\mathrm{of}\;\mathrm{control}-A\;\mathrm{amount}\;\mathrm{of}\;\mathrm{sample}}{A\;\mathrm{amount}\;\mathrm{of}\;\mathrm{control}}\right)\times 100$$

### 2.9. Samples Preparation of the Dried Oysters

The thawed oysters were boiled in boiling water for 3 min, the surface water was drained, and the oysters were then stored in the refrigerator (4 °C) until further processing. The oyster samples were randomly divided into two groups: (i) uncoated (Control); (ii) coated with RO nanoliposome solution (0.5 mg/mL), and the oysters were immersed in the RO nanoliposome solutions for 2 min, and got uniform coating on the surface. The uncoated and coated samples were dried at 50 °C for 8 h to obtain dried oysters. The samples of each group were packaged and sealed hermetically and stored at room temperature (25–28 °C) for 28 d.

### 2.10. Determination of Oxidative Stability of Dried Oysters

The oxidative stability of the dried oysters was characterized by carbonyl, sulfydryl content, POV, TBARs, and the content of 4-HNE, and determined at 1, 7, 14, 21, and 28 d. Each sample was run in triplicate.

Myofibril protein (MP) was isolated, as described by Xiong et al. [37]. MP was extracted with 4 vol (m/V) 10 mmol/L PBS (pH 7.0), containing 0.1 mol/L NaCl, 2 mmol/L MgCl_2_, 1 mmol/L EGTA, centrifugated, repeated 3 times. Then, the PBS pH was adjusted to 6 and the MP was extracted. Finally, the precipitate was dissolved in the 10 mmol/L PBS (pH 6.0) containing 0.6 mol/L NaCl, centrifugated, and the MP was then collected. The carbonyl content of dried oysters was evaluated by derivatization with 2,4-dinitrophenylhydrazine (DNPH) referring to Cao et al. [38], with some modifications. Briefly, 200 µL 5 mg/mL of MP was dissolvedin20 mM sodium PBS (pH 6.0). The solution was mixed with 800 µL 10 mM DNPH and incubated for 1 h (room temperature). The reacted sample was centrifuged at 8000× *g* for 5 min following the addition of 1 mL of 10% TCA to obtain the precipitate, which was washed 3 times with 1 mL of ethanol: ethyl acetate (1:1, *v*/*v*). The final precipitate was dissolved in 1.5 mL of 6 M guanidine HCl with PBS (pH 6.0, 20 mM). The solution mixture was centrifuged at 8000× *g* for 8 min. Carbonyl content was calculated by absorption in 370 nm. The content of MP was calculated at 280 nm by using BSA in 6 M guanidine HCl as a control. 

The sulfhydryl content of dried oysters was referring to Li et al. [39], 0.25 mL protein sample (2 mg/mL MP) was mixed with 50 μL 10 mM 5,5′-Dithiobis-(2-nitrobenzoic acid) (DTNB), 2.5 mL of 8 M urea, 10 mM ethylene-diaminetetraacetic acid (EDTA), 2 % sodium dodecyl sulfate (SDS) and 0.2 M Tris-HCl (pH 7.1), reacted at 40 °C for 15 min, and measured at 412 nm. 

The peroxide value (POV) of dried oysters was determined referring to the Liu et al. [40], and expressed as g/100 g lipid. To obtained total lipid, crushed oyster samples were combined with anhydrous ether (1:2–3, *w*/*v*). After shaking, soaked for more than 12 h, filtered with anhydrous sodium sulfate. The filtrate was and removed by rotary evaporator (40 °C), the residue is the total lipid to be tested. 0.20 g total lipid was dissolved in 50 mL of mixture of isooctane and glacial acetic acid (2:3; *v*/*v*). 1 mL saturated potassium iodide was added and stirred, then 30 mL deionized water was added, then determined by automatic potentiometric titrator.

The thiobarbituric acid reactive substances (TBARs) of dried oysters were detected referring to Pabast, et al. [24], with some modifications. 0.5 g sample was placed in 30 mL of perchloric acid solution (PCA, 4%) and 0.5 mL of butylated hydroxytoluene (BHT, 7.2%), homogenized at 8000 rpm for 30 s, and filtered through a filter. Then, mixed with 0.02 M TBA solution (1:1, *v*/*v*), reacted at 90 °C, and measured at 532 nm. The values were represented by mg malonaldehyde (MDA)/kg dried oysters.

The sample 4-HNE levels were detected by ELISA kit. Briefly, the standard was diluted on the ELISA-coated plate. The samples were homogenized with a given amount of PBS (pH 7.4). Samples were centrifuged at 2500 rpm for 20 min. The 10 μL supernatant and 40 μL sample diluent were added to the well. The sample was diluted and incubated at 37 °C for 30 min. Enzyme labeling reagent was added to each well, then incubated and washed with a washing solution. The chromogenic reagent was added, and the reaction was terminated after 15 min. The absorbance value of each well was measured at 450 nm with a microplate reader. 

### 2.11. Statistical Analysis

The DSC results were obtained by Universal Analysis V1.7F software. Data are presented as means ± standard deviations. The variance analysis and significant differences were used by Duncan’s multiple range test (SPSS 22.0, Armonk, NY, USA: IBM Corp.). 

## 3. Results and Discussion

### 3.1. Characteristics of Nanoliposomes and Encapsulation Efficiency

The particle size of nanoliposomes is important to their properties, functionalities, and stability under storage, transportation, and processing conditions. Nanoliposomes in the moderate size range (50–200 nm) will have better stability and utilization ratio [41,42]. Table 1 demonstrates that the average size of nanoliposomes increased significantly (*p* < 0.05) because of RO coating. The average sizes of empty and RO-coated nanoliposomes were 95.31 nm and 160.48 to 162.65 nm. However, the average sizes of nanoliposomes coated with different RO concentrations were not significantly different. The particle size of the coated nanoliposomes could be affected by the material type and preparation methods, such as ultrasonic dispersion and thin-film evaporation [43,44]. The PDI values give an indication of a particle size distribution in the emulsion, and the lower value showed higher stability [45]. The PDI values of empty nanoliposomes and different concentrations of RO nanoliposomes were between 0.266–0.283, and the different coated concentrations of RO did not significantly affect the PDI values of the nanoliposomes. The result indicated that the particle size of the nanoliposomes was well controlled.

Zeta potential is a potential difference between the surface of a nanoparticle and the ambient solution, and is a key physical variable used to evaluate a nanoparticle’s surface charge character, which indicates the physical stability of nanoparticles in liquids [46]. Table 1 shows that the prepared nanoliposomes had a negative zeta potential. The zeta potential of empty nanoliposomes was −14.27 mV. Compared to the empty nanoliposomes, the zeta potential of RO-coated nanoliposomes was significantly decreased (*p* < 0.05). The zeta potential decreased with the increase of added RO concentration (13–52 mg in 100 mg empty nanoliposome), but was not significant (*p* > 0.05). The particles can aggregate less and are more stable in solution if they have a larger absolute value of the zeta potential [47]. 

Encapsulation efficiency represents the ability to encapsulate the bioactive compounds in the nanoliposomes [48]. The encapsulation efficiency of different concentrations of RO are shown in Table 1. With an increase of RO from 13 to 26 mg, the encapsulation efficiency increased from 59.57% to 68.25%. However, when the concentration of RO reached 52 mg, the encapsulation efficiency decreased to 52.77%, which may be due to the limited volume of nanoliposomes; this finding agreed with previous reports [49,50,51]. Based on the above analysis, the concentration of 26 mg RO, which had optimal characteristics, was selected for preparing RO nanoliposomes for further experiments. 

### 3.2. FTIR Spectroscopy Analysis of Nanoliposomes

The chemical structure and functional groups of RO, empty nanoliposomes, and RO nanoliposomes were analyzed to determine by FTIR spectra in Figure 1. The spectra of RO revealed several typical peaks around 721 cm^−1^ (deformation vibration of C-H bond in benzene ring), 1160 cm^−1^ (deformation vibrations of O-H in-plane bending for phenol), 1502 cm^−1^ (stretching vibrations of benzene, pyridine, and other heteroaromatic rings), 1761 cm^−1^ (carbonyl C=O stretching at the ester bond), and 2859 cm^−1^ (alkyl C-H stretching vibration) [52,53]. These represent characteristic absorption peaks of functional groups in RO, which contain many polyphenolic compounds, including carnosic and rosmarinic acid [54]. At the wavenumbers 3462, 2859, 2832, 1748, 1080, 861, and 507 cm^−1^, the characteristic peak values of nanoliposomes containing RO and empty nanoliposomes were identical. This suggests that there was no chemical interaction between the bioactive and carrier compounds. There were no new characteristic peaks in this research, which indicated that no functional bond between the bioactive and carrier compounds was formed. All outcomes reflected that the RO and nanoliposomes were mixed physically and did not react chemically, suggesting the retention of antioxidant properties [51].

### 3.3. TEM of Nanoliposomes

TEM imagines were used for evaluating the particle size and shape of the nanoliposomes [55]. The micrograph showed that the nanoliposome particles were spherical and relatively dispersed because of the high zeta potential (Figure 2). The nanoliposomes became irregularly spherical in shape and increased in particle size after encapsulating RO (Figure 2b), which was similar to the findings of Nagahama et al. [56]. The particle size of empty nanoliposomes is about 100 nm, however, it was between 100 and 200 nm after RO coating. The result is similar to the mean particle size shown in Table 1. Similar TEM image results were also found in nisin-GE liposomes [57]. 

### 3.4. DSC Analysis of Nanoliposomes

DSC was performed for properties testing and quality control of the nanoliposomes [32]. The temperature range of DSC thermograms was 0–300 °C (Figure 3). In the case of RO, the peak temperature (Tm) was 149.8 °C. However, the Tm value of the encapsulated RO reached 234.0 °C, and the melting endotherm of RO disappeared. The changes in thermal properties suggest that RO inserts into the lipid bilayer of nanoliposomes and interacts with phospholipid molecules. It confirms that RO nanoliposomes improved the thermal stability of traditional RO. Similar findings were reported [32,58].

### 3.5. Temperatures and pH Stability of RO Nanoliposomes

As shown in Figure 4, the stability of RO nanoliposomes at different storage temperatures and pH were determined. The average sizes of RO nanoliposomes at 4 °C and 25 °C were 152.00–162.32 nm and 155.6–169.17 nm, respectively, during the storage time of 7 d (Figure 4a). Although the average size of the RO nanoliposome fluctuated during storage time, the RO nanoliposomes were relatively stable, and there was no significant difference between 7 d and 0 d (*p* > 0.05), which demonstrates that the aggregation or degradation might not have occured in nanoliposomes at different storage temperatures. Compared with 25 °C, nanoliposomes have higher stability under low temperature storage (4 °C), which is similar to the findings of Gülseren et al. [59].

The average sizes of RO nanoliposomes were measured at different pH values (2–12). As shown in Figure 4b, the average size of RO nanoliposomes reached the maximum value of 160.37 nm at pH 7 (*p* < 0.05), and decreased slightly with the increasing of acidity (pH 1–6) or alkalinity (pH 8–12), which revealed the stability of RO nanoliposomes in different pH environments, and showed that the nanoliposomes had potential for application in various foods. However, our results were inconsistent with those of Cheng et al. [60], who found the nanoliposomes had higher average size at pH, which might be caused by the composition of the nanoliposomes. 

### 3.6. Radical-Scavenging Activity of RO Nanoliposomes during Storage

RO contains the active compounds, such as phenol, terpene, and flavonoids which have been proved to be effective antioxidants [61]. DPPH radical-scavenging activity and ABTS radical-scavenging activity are critical indexes to reflect the antioxidant activity of natural compounds [62]. We compared the antioxidant activities of RO and RO nanoliposomes during storage. The DPPH radical-scavenging activity of RO decreased from 72% to 13% during 4 weeks of storage (*p* < 0.05) (Figure 4a). After encapsulation by nanoliposomes, the DPPH radical-scavenging activity decreased slightly from 56% to 48% (*p* < 0.05), which indicated the strong DPPH radical-scavenging rate of RO. The ABTS radical-scavenging activity of RO and RO coated by nanoliposomes was decreased from 65% to 16% (*p* < 0.05) and 56% to 45% (*p* < 0.05), respectively, during 4 weeks’ storage, as presented in Figure 4b. The trend was similar to that of DPPH radical-scavenging activity. The result showed that RO showed better stability and decreased the loss of antioxidant activity during storage time after encapsulation by nanoliposomes, and that encapsulation can preserve RO decomposition. At the beginning of storage, the DPPH and ABTS radical-scavenging activity of RO was higher than RO nanoliposomes, possibly because some compounds in RO are bonded to active groups, such as phospholipids, thus reducing the chances of reacting with DPPH free radicals [63]. By encapsulation, active ingredients are effectively protected during storage [64].

### 3.7. Effect of RO Nanoliposomes on Oxidative Stability of Dried Oysters during Storage

#### 3.7.1. Lipid Oxidation of Dried Oysters

Dried oysters contain a certain amount of lipids in their composition. Lipid oxidation is a deterioration reaction that not only reduces the nutritional value, but also can lead to the generation of harmful compounds [65,66,67]. The POV of the control and RO nanoliposomes group were shown in Figure 5a. The POV of the two groups were noted to increase steadily, indicating that lipid oxidation occurred over the course of the entire storage time of the dried oysters, due to the formation of primary lipid oxidation products [68]. It can be seen that the POV of the control showed a slight increase until 14 days of storage. Thereafter, the POV increased sharply from 0.20 g/100 g lipid to 0.42 g/100 g lipid in dried oysters preserved for 28 d (*p* < 0.05). The value increased more than five times over the course of the entire storage. However, the POV of RO nanoliposomes group showed a slowly increasing tendency, compared with the control. The POV value decreased by half of the control at 28 d of storage (*p* < 0.05). Obviously, lipid oxidation was inhibited in the oysters of the RO nanoliposomes-treated group. Similarly, Wu et al. [69] also found that the POV of the nanoliposome samples embedded with antioxidant active ingredients had a significant reduction during storage.

The TBARs value reflects the result of the derivatives of oxidation of unsaturated fatty acids reacted with TBA during animal fat storage, which is generally used to measure the formation of secondary oxidation products. As Figure 5b shows, the level of TBARs increases as lipid oxidation progresses, which may be due to the high content of PUFA in oysters [70]. Lipid oxidation of dried oysters increased over the course of the entire storage period, and the control always showed higher oxidation than the RO nanoliposomes. At the end of storage, the TBARs value reached 1.2 mg/kg in the control compared with 0.9 mg/kg of the RO nanoliposomes, which was 25% lower (*p* < 0.05). The result showed that RO nanoliposomes effectively inhibited lipid oxidation of dried oysters during storage (*p* < 0.05), which was due to its sustainable antioxidant capacity in the storage of dried oysters. Nanoliposome encapsulation can better protect the antioxidant components of RO, like carnosic acid, carnosol, and α-pinene [71]. Furthermore, the inhibition effect of RO in secondary lipid oxidation also has been confirmed in aquatic products, such as fish [72] and shrimp [73].

#### 3.7.2. Protein Oxidation of Dried Oysters

Protein oxidation occurs frequently during the processes of cold air-drying and storage. Carbonyls resulted in the base residues of amino acids oxidized by oxygen and light, which can represent the extent of protein oxidation in food. As we can see in Figure 6, with a prolonged storage time, the carbonyl content of dried oysters increased simultaneously; meanwhile, protein oxidation occured constantly. The carbonyl content of RO nanoliposomes increased from 0.9 nmol/mg to 3.0 nmol/mg in 28 d, compared with the control, which increased from 0.9 nmol/mg to 4.9 nmol/mg, which means that protein oxidation can be inhibited by adding nanoliposomes during storage (*p* < 0.05). This may be due to the interaction of RO nanoliposomes and dried oysters and may increase the specific surface area of the nano-size RO liposomes [74]. The inhibitory effect of rosemary essential oil on protein oxidation was also found in shrimp [75,76]. Nanoliposome encapsulation can be used as an effective method to inhibit protein oxidation in aquatic products. Cui et al. [77] found that nano-encapsulated Litseacubeba essential oil (LC-EO) can delay the protein oxidation of salmon.

The content of sulfhydryl is also an important indicator of protein oxidation, and plays an important role in the structure and function of proteins. As shown in Figure 7, with the increase of storage time, the sulfhydryl content of dried oysters decreased significantly (*p* < 0.05). The lower the sulfhydryl content, usually the higher the protein oxidation, as the sulfhydryl were easily oxidized into disulfide bonds by free radicals [78]. In Figure 7, the sulfhydryl content of 0 d to 28 d in control and RO nanoliposomes were 173.89 to 113.93 μmol/g protein and 171.26 to 136.23 μmol/g protein, respectively, which further indicated that protein oxidation occurred in dried oysters during storage. Similar results were found in the low temperature oysters [79], and catfish [80]. Moreover, the sulfhydryl contents in the control group were lower than in the RO nanoliposome group, which indicated that RO nanoliposomes could inhibit the reduction of sulfhydryl content to inhibit protein oxidation during the storage of dried oysters.

#### 3.7.3. 4-HNE Content Analysis of Dried Oysters

4-HNE is a toxic aldehyde and a lipid peroxidation end product derived mainly from n-6 PUFA [81]. Changes in the 4-HNE content of dried oysters during 28 d of storage are presented in Figure 8. The 4-HNE content of dried oysters was increased during storage. However, during the storage period, the oxidation products in dried oysters coated with RO nanoliposomes were (*p* < 0.05) lower than the control, which validated our findings reported for radical scavenging, lipid peroxidation, etc. After two weeks (14 d), in dried oysters, 4-HNE increased faster than before in the RO nanoliposome group, which showed that RO nanoliposomes have a stronger protective effect in peroxidation of n-6 PUFA from 0 to 14 d [82]. The toxicity of 4-HNE was related to its content in food. A measure of 4-HNE content has been reported in several seafoods and meats, like frozen fish (0.015–0.75 µmol/mg) [83], fried clams (0.41-1.43 μg/g) [84], and pork meat (27.96 nmol/g) [85]. In our study, the content of 4-HNE in dried oysters was relatively lower (16.8–32.0 ng/kg), which may be due to the lower content of n-6 PUFA. Asha et al. [86] found that n-3 fatty acids took up a larger proportion in oysters. Moreover, 4-HNE can interact with amino acids in the protein, which may also cause lower content of 4-HNE [87]. However, the interaction of 4-HNE and protein needs to be further studied.

## 4. Conclusions

In this study, nanoencapsulation was applied to RO, and their physicochemical characteristics and their application for the oxidative stability of dried oysters were observed. RO nanoliposomes were obtained by the thin-film evaporation method, which showed good stability. RO nanoliposomes had better thermal stability properties in comparison with RO. The coated-RO nanoliposomes showed sustained release effects and better antioxidant stability. RO nanoliposomes could be used as natural antioxidants for dried oysters, improving their stability, and preventing protein and lipid oxidation. Better antioxidant stability during storage and the thermal processing properties of RO nanoliposomes can help prolong food shelf life. The results of our study revealed that RO nanoliposomes have good application prospects in the arena of food processing.

## Figures and Tables

**Figure 1 bioengineering-09-00818-f001:**
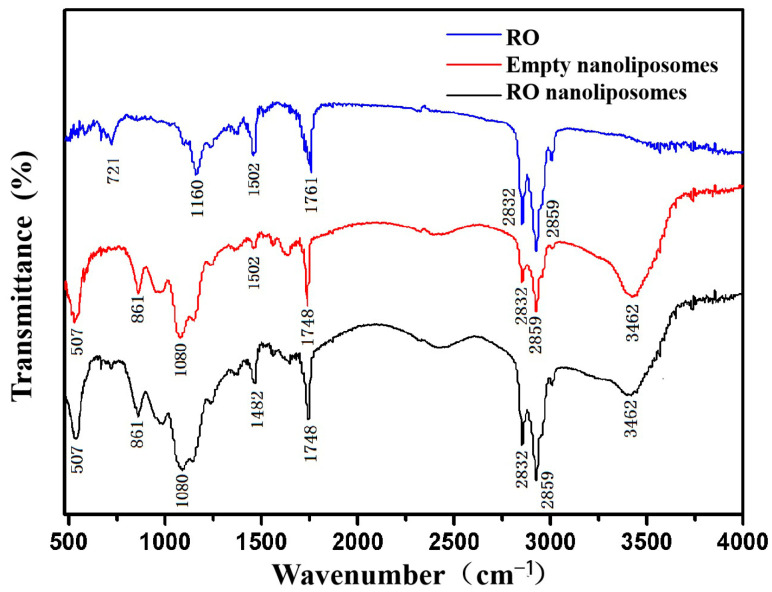
FTIR spectra of RO, empty nanoliposomes, and RO nanoliposomes.

**Figure 2 bioengineering-09-00818-f002:**
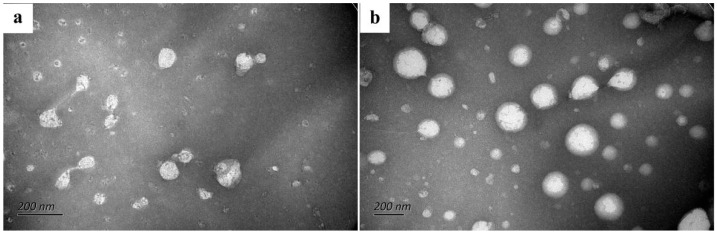
TEM images of (**a**) (empty nanoliposomes) and (**b**) (RO nanoliposomes).

**Figure 3 bioengineering-09-00818-f003:**
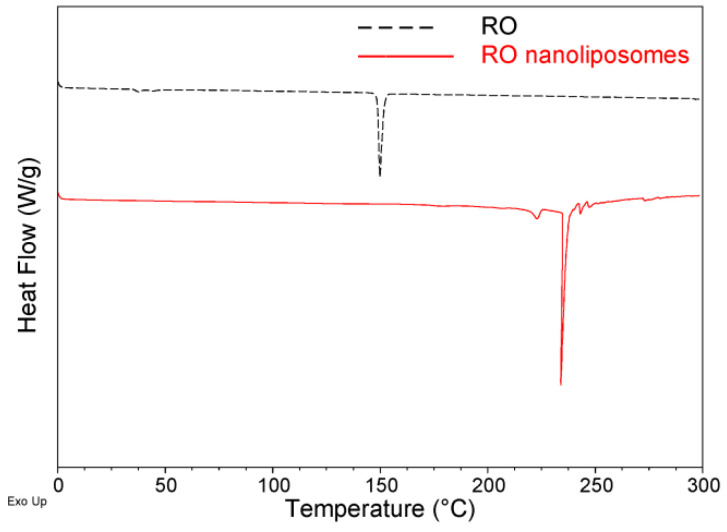
Thermal behavior of RO and RO nanoliposomes by DSC.

**Figure 4 bioengineering-09-00818-f004:**
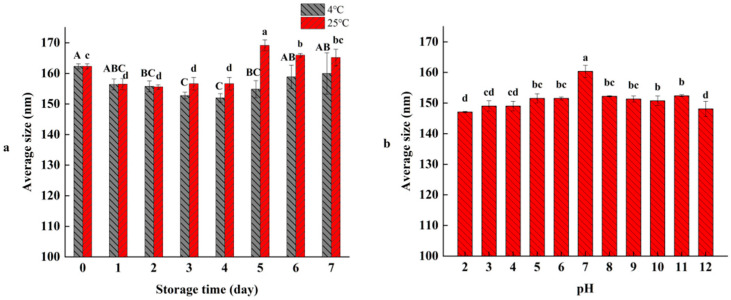
Effect of temperature (**a**) and pH (**b**) on average size of RO nanoliposomes. Different case letters (a–d; A–C) in each panel indicate significant difference from each other (*p* < 0.05).

**Figure 5 bioengineering-09-00818-f005:**
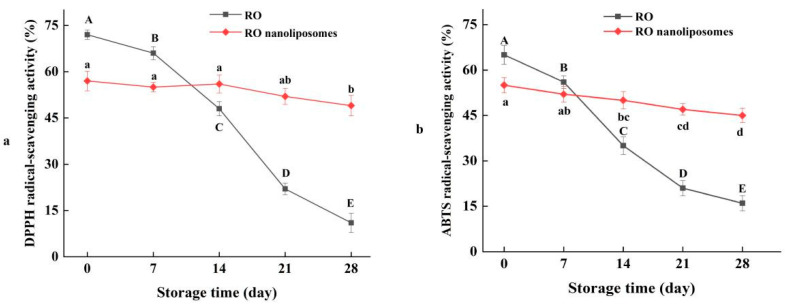
(**a**) DPPH radical-scavenging activity of RO and RO nanoliposomes during storage; (**b**) ABTS radical-scavenging activity of RO and RO nanoliposomes during storage. Different case letters (a–d; A–E) in each line indicate significant difference from each other (*p* < 0.05).

**Figure 6 bioengineering-09-00818-f006:**
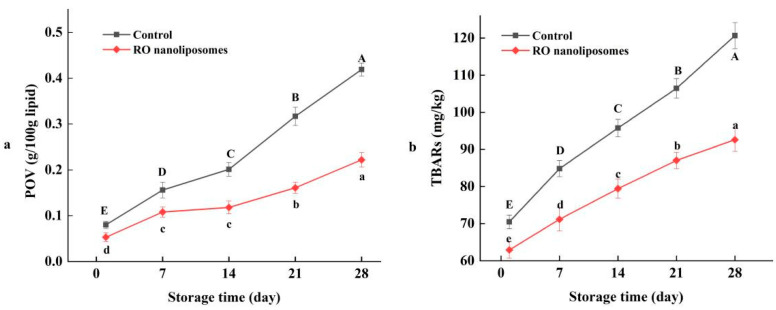
(**a**) POV of dried oysters during storage; (**b**) TBARs of dried oysters during storage. Different case letters (a–e; A–E) in each line indicate significant difference from each other (*p* < 0.05).

**Figure 7 bioengineering-09-00818-f007:**
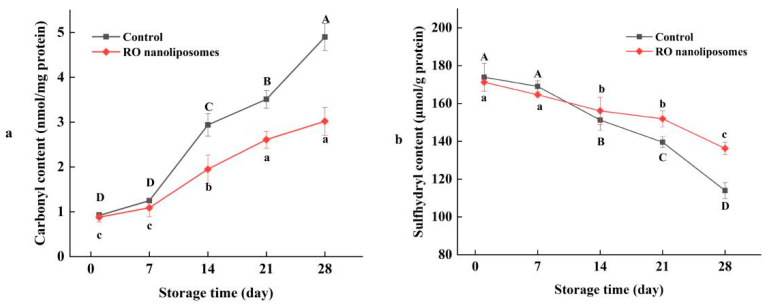
Carbonyl (**a**) and sulfhydryl (**b**) content of dried oysters during storage. Different case letters (a–c; A–D) in each line indicate significant difference from each other (*p* < 0.05).

**Figure 8 bioengineering-09-00818-f008:**
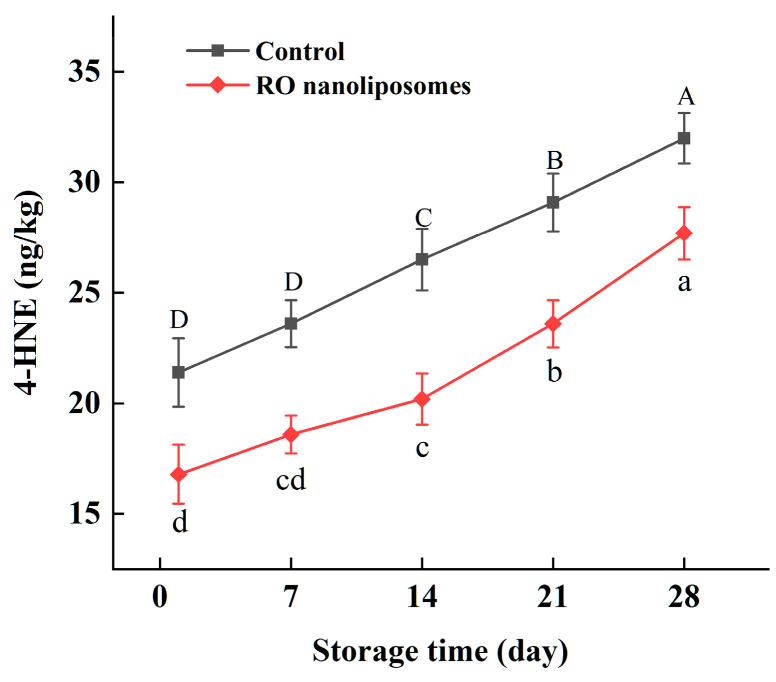
4-HNE of dried oysters during storage. Different case letters (a–d; A–D) in each line indicate significant difference from each other (*p* < 0.05).

**Table 1 bioengineering-09-00818-t001:** Mean particle size, PDI, and zeta potential of nanoliposomes loaded with different content of RO.

Treatment	Average Size (nm)	PDI	Zeta Potential (mV)	Encapsulation Efficiency (%)
Empty nanoliposome	95.31 ± 1.02 ^a^	0.266 ± 0.005 ^a^	−14.27 ± 0.32 ^a^	-
Coated 13 mg RO	160.48 ± 0.95 ^b^	0.281 ± 0.005 ^a^	−15.37 ± 0.21 ^b^	59.57 ± 1.02 ^a^
Coated 26 mg RO	162.32 ± 0.86 ^b^	0.283 ± 0.006 ^a^	−16.03 ± 0.28 ^c^	68.25 ± 2.38 ^b^
Coated 39 mg RO	162.65 ± 0.96 ^b^	0.269 ± 0.008 ^a^	−16.07 ± 0.33 ^c^	55.69 ± 1.96 ^c^
Coated 52 mg RO	162.25 ± 1.21 ^b^	0.266 ± 0.007 ^a^	−16.30 ± 0.12 ^c^	52.77 ± 2.89 ^d^

Different lowercase letters (a, b, c, d) in the same column indicated significant difference (*p* < 0.05) of means for each measurement.

## Data Availability

The data presented in this study are available on request from the corresponding author.

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
