# Peer review of "Physicochemical and Antioxidant Properties of Nanoliposomes Loaded with Rosemary Oleoresin and Their Oxidative Stability Application in Dried Oysters"

_bioengineering, 2022, doi:10.3390/bioengineering9120818_

Round 1
Reviewer 1 Report
This is an interesting study about the physicochemical and antioxidant properties of nanoliposomes. I suggest it for publication after the following minor points are solved.
1. The loading content should be determined.
2. How is the physical stability of the liposomes?
3. What's the difference between liposomes and nanoliposomes?
4. Figure 2, it seems they are micelles but not liposomes.
5. Line 61-62, several studies (doi.org/10.1016/j.actbio.2021.02.001; doi.org/10.1021/acs.langmuir.8b00851) should be included to support such a claim.
Author Response
Dear Review,
Please find attached the responses. We are grateful to the reviewers for the comments, which have significantly improved the quality of the manuscript. We have made our best effort to change the manuscript accordingly. Please find below our answers to the reviewers’ comments. In the manuscript, the required changes have been marked red. We hope that the changes made to the manuscript and our attached explanations will be sufficient to make it acceptable for publication in Bioengineering.
We shall look forward to hearing from you at your earliest convenience.
With kind regards
Prof. Mingwu Zang
China Meat Research Center, Beijing Academy of Food Sciences
E-mail: [email protected] (M. W. Zang)
Telephone: +86-010-67282165
We would like to express our sincere thanks to the reviewer’s constructive and positive comments.
Responses to the comments of Reviewer
Reviewer #1:
- The loading content should be determined.
Response: Thank you for pointing this out. The loading content of RO was 26 mg for preparing RO nanoliposomes in experiments, see lines 281, Section “3.1. Characteristics of nanoliposomes and encapsulation efficiency”. RO was purchased from Kalsec Co, Ltd (Shanghai, China), and the content of carnosic acid was 15.36%, (w/w), see lines 105, carnosic acid is one of the main antioxidant ingredients in RO, the content of carnosic acid was measured to evaluate the antioxidant capacity of RO, and the encapsulation efficiency was calculated by carnosic acid content.
Encapsulation efficiency represents the ability to encapsulate the bioactive compounds in the nanoliposomes [50]. The encapsulation efficiency of different concentrations of RO showed in Table 1. With the increasing RO from 13 to 26 mg, the encapsulation efficiency increased from 59.57% to 68.25%. However, when the concentration of RO reached 52 mg, the encapsulation efficiency decreased to 52.77% which may be due to the limited volume of nanoliposomes, this finding agreed with previous reports [51, 52, 53]. Based on the above analysis, the concentration of 26 mg RO which had optimal characteristics, was selected for preparing RO nanoliposomes for further experiments.
RO (the content of carnosic acid was 15.36%, w/w) was obtained from Kalsec Co, Ltd (Shanghai, China).
- How is the physical stability of the liposomes?
Response: RO nanoliposomes is stable under different storage temperatures and pH conditions. we have added the stability of RO nanoliposomes in the manuscript, see lines, in Section “3.5 Temperatures and pH stability of RO nanoliposomes”
3.5 Temperatures and pH stability of RO nanoliposomes
As shown in Fig. 4, the stability of RO nanoliposomes at different storage temperatures and pH were determined. The average sizes of RO nanoliposomes at 4 °C and 25 °C were 152.00–162.32 nm and 155.6–169.17 nm respectively, during the storage time of 7 d (Fig. 4a). Although the average size of the RO nanoliposome fluctuated during storage time, the RO nanoliposomes were relatively stable, and there was no significant difference between 7 d and 0 d (p>0.05), which demonstrate the aggregation or degradation might not occured in nanoliposomes at different storage temperatures. Compared with 25 °C, nanoliposomes have higher stability under low temperature storage (4 °C), which were similar with Gülseren et al. [61].
The average size of RO nanoliposomes were measured at different pH values (2–12). As shown in Fig. 4b, the average size of RO nanoliposomes reaching the maximum value 160.37 nm at pH 7, and decreased slightly with the increasing of acidity (pH 1–6) or alkalinity (pH 8–12), revealed the stability of RO nanoliposomes in different pH environments, and the nanoliposomes had potential to application in various foods. However, our results were inconsistent with Cheng’s et al [62], who found the nanoliposomes had higher average size at pH, which might caused by the composition of nanoliposomes.
Figure 4. Effect of temperature (a) and pH (b) on average size of RO nanoliposomes.
- What's the difference between liposomes and nanoliposomes?
Response: Nanoliposome exclusively refer to nanoscale bilayer lipid vesicles, and liposome is a general terminology covering many classes of vesicles whose diameters range from tens of nanometers to several micrometers. In our study the average size of RO-coated nanoliposomes was 160.48 to 162.65 nm. Compared to liposomes, nanoliposomes provide more surface area and have the potential to increase solubility, enhance bioavailability, improve controlled release and enable precision targeting of the encapsulated material to a greater extent.
- Figure 2, it seems they are micelles but not liposomes.
Response: We have revised the figure 2.
Figure 2. TEM images of a (empty nanoliposomes) and b (RO nanoliposomes).
- Line 61-62, several studies (doi.org/10.1016/j.actbio.2021.02.001; doi.org/10.1021/acs.langmuir.8b00851) should be included to support such a claim.
Response: Thank you for pointing this out. We have added these studies in our revised manuscript, see lines 84.
Appropriate encapsulation of bioactives with micrometric-size capsules can improve the controlled release properties, and in general, does not affect the activities [20, 21, 22, 23].
- Zhang, M., Hagan IV, C. T., Foley, H., Tian, X., Yang, F., Au, K. M.,et al. (2021). Co-delivery of etoposide and cisplatin in dual-drug loaded nanoparticles synergistically improves chemoradiotherapy in non-small cell lung cancer models. Acta biomaterialia, 124, 327-335.https://doi-org.hy223.top/10.1016/j.actbio.2021.02.001.
23.Lin, W., Ma, G., Yuan, Z., Qian, H., Xu, L., Sidransky, E., & Chen, S. (2018). Development of zwitterionic polypeptide nanoformulation with high doxorubicin loading content for targeted drug delivery. Langmuir, 35(5), 1273-1283. https://doi.org/10.1021/acs.langmuir.8b00851.

Reviewer 2 Report
The present manuscript focuses on the hysicochemical and antioxidant properties of nanoliposomes loaded with rosemary oleoresin and their oxidative stability application in dried oysters. The subject frame of the work is well constructed. So, in this respect and this article should be contributed to present research. I recommended this work for publication after the following minor revisions.
1. There are several typographical mistakes as well in whole manuscript. Therefore, the author’s thoroughly careful check the language and typo mistake to minimize the error.
2. The abstract should be beginning with a sentence about the background of concept and the aims as well as novelty of study should be mentions. What exactly is the novelty of this study? The abstract is poorly written and should be improved. Abbreviations must be avoided in abstract. Parenthesis should be avoided in abstract - this is poor writing. Please improve.
3. The introduction and discussion section need extensive revision and improved. Be specific and adhere to importance of topic.
4. All figures are of poor technical quality and not suitable for publication, especially in a high reputed journal. Font size and kind is too small and must be unified in all figures. Small writings are unreadable. All figures must be self-explanatory. Axis titles are poorly presented or absent. Units are missing. Are the data presented in figures significantly different? At least error bars should be shown.
5. What is exactly the novelty of this review article, as so many articles were already out, is this the updates version or some other novel aspect. Author needs to revised it carefully and should provide novelty statement.
Author Response
Dear Review,
Please find attached the responses. We are grateful to the reviewers for the comments, which have significantly improved the quality of the manuscript. We have made our best effort to change the manuscript accordingly. Please find below our answers to the reviewers’ comments. In the manuscript, the required changes have been marked red. We hope that the changes made to the manuscript and our attached explanations will be sufficient to make it acceptable for publication in Bioengineering.
We shall look forward to hearing from you at your earliest convenience.
With kind regards
Prof. Mingwu Zang
China Meat Research Center, Beijing Academy of Food Sciences
E-mail: [email protected] (M. W. Zang)
Telephone: +86-010-67282165
We would like to express our sincere thanks to the reviewer’s constructive and positive comments.
Responses to the comments of Reviewer
Reviewer #2:
- There are several typographical mistakes as well in whole manuscript. Therefore, the author’s thoroughly careful check the language and typo mistake to minimize the error.
Response: We have revised the whole manuscript carefully and tried to avoid any language and typo mistake to minimize the error.
- The abstract should be beginning with a sentence about the background of concept and the aims as well as novelty of study should be mentions. What exactly is the novelty of this study? The abstract is poorly written and should be improved. Abbreviations must be avoided in abstract. Parenthesis should be avoided in abstract - this is poor writing. Please improve.
Response: Thank you for pointing this out, we have revised the abstract to avoided abbreviations, and added the novelty of this study, lines 11 in section “Abstract”.
Abstract: Lipid and protein oxidation is a main problem related to dried aquatic products preservation. Rosemary oleoresin is widely used as an antioxidant, but its application is limited due to the instability and easy degradation. Nanoliposome encapsulation is a promising and rapidly emerging technology in which antioxidants are incorporated into the liposomes to provide the food high quality, safety and long shelf life. The objectives of this study were to prepare nanoliposomes coating of rosemary oleoresin to enhance the antioxidant stability, and their potential application in inhibiting protein and lipid oxidation in dried oysters during storage. The nanoliposomes encapsulating rosemary oleoresin were used by a thin-film evaporation method, and the optimal amount of encapsulated rosemary oleoresin was chosen based on changes in the dynamic light scattering, Zeta potential, and encapsulation efficiency of the nanoliposomes. The fourier transform-infrared spectroscopy of rosemary oleoresin nanoliposomes showed no new characteristic peaks formed after rosemary oleoresin encapsulation, and the particle size of rosemary oleoresin nanoliposomes was 100–200 nm in transmission electron microscopy. The differential scanning calorimetry indicated that the nanoliposomes coated with rosemary oleoresin had better thermal stability. Rosemary oleoresin nanoliposomes presented good antioxidant stability, and still maintained 48% DPPH radical-scavenging activity and 45% ABTS radical-scavenging activity after 28 d of storage, which was 3.7 times and 2.8 times higher than that of empty nanoliposomes, respectively. Compared with the control, the dried oysters coated with rosemary oleoresin nanoliposome showed significantly lower values of carbonyl, sulfhydryl content, thiobarbituric acid reactive substances, Peroxide value, and 4-Hydroxynonenal contents during 28 d of storage. The results provide a theoretical basis for developing an efficient and long-term antioxidant approach.
- The introduction and discussion section need extensive revision and improved. Be specific and adhere to importance of topic.
Response: We have revised the introduction and discussion section to specific and adhere the importance of the topic.
- All figures are of poor technical quality and not suitable for publication, especially in a high reputed journal. Font size and kind is too small and must be unified in all figures. Small writings are unreadable. All figures must be self-explanatory. Axis titles are poorly presented or absent. Units are missing. Are the data presented in figures significantly different? At least error bars should be shown.
Response: We have replaced them with more clear pictures, unified the font size, and added the error bars, see Figre1-8.
- What is exactly the novelty of this review article, as so many articles were already out, is this the updates version or some other novel aspect. Author needs to revised it carefully and should provide novelty statement.
Response: Thank you for pointing this out, the objectives of this study were to prepare nanoliposomes coating of RO to enhance the antioxidant stability, and their potential application in inhibiting protein and lipid oxidation in dried oysters during storage. We have strengthened the expression of relevant contents, and revised the introduction carefully and provide novelty statement, see lines 64, and 90.
Rosemary spices is a potential functional ingredient of meat products,like beef, chicken and pork,due to the excellent antioxidant and antibacterial properties [16]. Rosemary spice has many forms, such as rosemary essential oil, rosemary extract, rosemary resin, and ground [17]. Oleoresins are viscous liquid extracted from spices, which contain not only essential oil but also some non-volatile components, such as pigment, fatty oils and phenolic antioxidants. Compared to essential oils, the oleoresins possibly are able to provide sustainable antioxidant capacity due to their low volatility. Therefore, in this study rosemary oleoresin (RO) was used in the nanoliposome encapsulation. Rosemary Oleoresins, as an active ingredient, are viscous liquid extracted from spices, which had lower volatility and long anti-oxidation time than essential oils and extracts [18]. However, plant oleoresins are sensitive to temperature, air, light, and relative humidity, resulting in chemical instability and easy degradation. Some bioactive ingredients of plant oleoresins are highly hydrophobic and poorly water-soluble, reducing bioavailability and restricting their application [19].
Additionally, the current research focused more on the chemical and physiological characteristics of nanoliposomes [28, 29, 30, 31, 32, 33], rather than the application in food. Furthermore, no data exist on the application of the edible coating containing RO-loaded nanoliposomes in dried oysters.

Reviewer 3 Report
Overall the manuscript is ok. However, the author needs to address the following issues.
1. The introduction should be improved with the importance of the current research.
2. The nanoparticle characterization should be carried out for EDX, particle size analysis, XRD.
3. The TEM image is not clear, the author should take the image at 5 and 10 nm.
4. Only DPPH and ABTS assay is not sufficient for antioxidant potential evaluation, the author should carry out more assay such as lipid peroxidation and GST, SOD inhibition assays.
5. The author should mention clearly from where, and during which season they obtained the oysters.
6. Proper identification of the sample in the study should be clarified.
7. Whether ethical approval has been taken for animal study should be mentioned.
8. Detailed methodology of "determination of oxidative stability of dried oyster should be described.
Author Response
Dear Review,
Please find attached the responses. We are grateful to the reviewers for the comments, which have significantly improved the quality of the manuscript. We have made our best effort to change the manuscript accordingly. Please find below our answers to the reviewers’ comments. In the manuscript, the required changes have been marked red. We hope that the changes made to the manuscript and our attached explanations will be sufficient to make it acceptable for publication in Bioengineering.
We shall look forward to hearing from you at your earliest convenience.
With kind regards
Prof. Mingwu Zang
China Meat Research Center, Beijing Academy of Food Sciences
E-mail: [email protected] (M. W. Zang)
Telephone: +86-010-67282165
We would like to express our sincere thanks to the reviewer’s constructive and positive comments.
Responses to the comments of Reviewer
Reviewer #3:
- The introduction should be improved with the importance of the current research.
Response: Thank you for pointing this out, we have revised the introduction, see lines 84, Section, “Introduction”
To date, the characterization and activities of nanoliposome-coated essential oils, such as cinnamon essential oil, Satureja khuzestanica essential oil, artemisia annua oil, thyme (Thymus zygis) essential oil, as well as encapsulated clove and lavender essential oil and others, have been relatively more studied [24, 25, 26, 27]. However, comparatively little research is available about the characteristics and activities of nanoliposome-encapsulated plant oleoresins. Additionally, the current research focused more on the chemical and physiological characteristics of nanoliposomes [28, 29, 30, 31, 32, 33], rather than the application in food. Furthermore, no data exist on the application of the edible coating containing RO-loaded nanoliposomes in dried oysters.
- The nanoparticle characterization should be carried out for EDX, particle size analysis, XRD.
Response: Thank you for pointing this out. The particle size was determined in the manuscript, see lines 250, Section “3.1. Characteristics of nanoliposomes and encapsulation efficiency”. EDX was used to determine the elements contained in the sample, like Ag+, and XRD was used to identify the phase in the sample (crystallization, crystal phase, crystal structure and bonding state, etc), referring to El Kurdi, and Patra, (2017) (DOI: 10.1039/c6cp08607a). However, we were more concerned about whether the antioxidant activity of RO was affected after encapsulation of nanoliposomes, so the FTIR spectroscopy of nanoliposomes was analysed in the manuscript. Thank you for your research ideas. In future research, we can explore the effects of adding different metal ions on the characterization and antioxidation of RO nanoliposomes by EDX and XRD.
3.1. Characteristics of nanoliposomes and encapsulation efficiency
The particle size of nanoliposomes is important to their properties, functionalities, and stability under storage, transportation, and processing conditions. Nanoliposomes in the moderate size range (50–200 nm) will have better stability, and utilization ratio [43, 44]. Table 1 demonstrates that the average size of nanoliposomes increased significantly (P < 0.05) because of coating RO. The average size of empty and RO-coated nanoliposomes was 95.31 nm and 160.48 to 162.65 nm. However, the average size of nanoliposomes coated with different RO concentrations were no significant differences. The particle size of the coated nanoliposomes could be affected by the material type and preparation methods, such as ultrasonic dispersion and thin-film evaporation [45, 46]. The PDI values give an indication of a particle size distribution in the emulsion, and the lower value showed higher stability [47]. The PDI values of empty nanoliposomes and different concentrations of RO nanoliposomes were between 0.266–0.283, and the different coated concentrations of RO do not affect significantly the PDI values of nanoliposomes. The result indicated that the particle size of the nanoliposomes was well controlled.
- The TEM image is not clear, the author should take the image at 5 and 10 nm.
Response: Thank you for pointing this out, we have replaced it with a clearer one. In the TEM image of nanoliposomes, 100 nm or 200 nm was usually used, caused the image at 5 and 10 nm might unable to accurately reflect the overall state of nanoliposomes, and the image is based on the following studies: Pinilla et al. (DIO:10.1016/j.ifset.2016.07.017), Colas, et al. (DIO: 10.1016/j.micron.2007.06.013), and Bianchi et al. (DIO: 10.3390/ijms21103436). In our study the average size of RO-coated nanoliposomes was 160.48 to 162.65 nm, so we took the image at 200 nm.
Figure 2. TEM images of a (empty nanoliposomes) and b (RO nanoliposomes).
- Only DPPH and ABTS assay is not sufficient for antioxidant potential evaluation, the author should carry out more assay such as lipid peroxidation and GST, SOD inhibition assays.
Response: Thank you for pointing this out, the content of GST and SOD in dried oyster were relatively lower, which were 72.0 ± 1.6 μmol/mg protein and 10.2 ± 3.5U/mg protein respectively (Neri et al., 2021; DIO: 10.1007/s10068-021-00907-x).
However, our study focused on the storage and application effect of nanoliposomes on dried oysters, the lipid peroxidation indexes (POV, TBARs, 4-HNE) in dried oysters were determined, and we did not pay too much attention to antioxidant enzymes activity. In the future research, we will strengthen the research on these contents to further clarify the antioxidant properties of RO nanoliposomes.
- The author should mention clearly from where, and during which season they obtained the oysters.
Response: Thank you for pointing this out, we have added the details of oysters, see lines 102, Section, “2.1. Materials”
The same batch of frozen oysters from South China Sea, with good commodity quality were obtained in Raoping Yujie Food Industry Co., Ltd. (China), transported to the laboratory through a cold chain, stored at –20°C, and transferred to 4 °C, the night before the experiment for defrosting.
- Proper identification of the sample in the study should be clarified.
Response: We have added the proper identification of oysters, see lines 102.
The same batch of frozen oysters from South China Sea.
- Whether ethical approval has been taken for animal study should be mentioned.
Response: Thank you for pointing this out, we studied the potential application in inhibiting protein and lipid oxidation in dried oysters during storage of RO nanoliposomes, and not involved the ethical approval from animal study.
- Detailed methodology of "determination of oxidative stability of dried oyster should be described.
Response: We have described the detailed methodology of "determination of oxidative stability of dried oysters”, see lines 195, Section, “2.10. Determination of oxidative stability of dried oysters”
2.10. Determination of oxidative stability of dried oysters
The oxidative stability of dried oysters was characterized by carbonyl, sulfydryl content, POV, TBARs, and the content of 4–HNE, and determined at 1, 7, 14, 21, and 28 d. Each sample was run in triplicate.
Myofibril protein (MP) was isolated as described by Xiong et al. [39]. MP was extracted with 4 vol (m/V) 10 mmol/L PBS (pH 7.0), containing 0.1 mol/L NaCl, 2 mmol/L MgCl2, 1 mmol/L EGTA, centrifugated, repeated 3 times. Then adjusted PBS pH to 6 and extracted. Finally, precipitate was dissolved in the 10 mmol/L PBS (pH 6.0), containing 0.6 mol/L NaCl, centrifugated, then the MP was collected. The carbonyl content of dried oysters was evaluated by derivatization with 2,4–dinitrophenylhydrazine (DNPH) referring to Cao et al. [40], with some modifications. Briefly, 200 µL 5 mg/mL of MP was dissolvedin20 mM sodium PBS (pH 6.0). The solution was mixed with 800 µL 10 mM DNPH and incubated for 1 h (room temperature). The reacted sample was centrifuged at 8,000 ×g for 5 min following the addition of 1 mL of 10% TCA to obtain the precipitate, which was washed 3 times with 1 mL of ethanol: ethyl acetate (1:1, v/v). The final precipitate was dissolved in 1.5 mL of 6 M guanidine HCl with PBS (pH 6.0, 20 mM). The solution mixture was centrifuged at 8,000 ×g for 8 min. Carbonyl content was calculated by absorption in 370 nm. The content of MP was calculated at 280 nm by using BSA in 6 M guanidine HCl as a control.
The sulfhydryl content of dried oysters was referring to Li et al. [41], 0.25 mL protein sample (2 mg/mL MP) was mixed with 50 μL 10 mM 5,5′–Dithiobis–(2–nitrobenzoic acid) (DTNB), 2.5 mL of 8 M urea, 10 mM ethylene–diaminetetraacetic acid (EDTA), 2 % sodium dodecyl sulfate (SDS) and 0.2 M Tris–HCl (pH 7.1), reacted at 40 °C for 15 min, and measured at 412 nm.
The peroxide value (POV) of dried oysters was determined referring to the Liu et al. [42], and expressed as g/100g lipid. To obtained total lipid, crushed oyster samples were combined with anhydrous ether (1:2–3, w/v). After shaking, soaked for more than 12 hours, filtered with anhydrous sodium sulfate. The filtrate was and removed by rotary evaporator (40 °C), the residue is the total lipid to be tested. 0.20 g total lipid was dissolved in 50 mL of mixture of isooctane and glacial acetic acid (2:3; v/v). 1 mL saturated potassium iodide was added and stirred, then 30 ml deionized water was added, then determined by automatic potentiometric titrator.
The thiobarbituric acid reactive substances (TBARs) of dried oysters were detected referring to Pabast, et al. [25], with some modifications. 0.5 g sample was placed in 30 mL of perchloric acid solution (PCA, 4%) and 0.5 mL of butylated hydroxytoluene (BHT, 7.2%), homogenized at 8000 rpm for 30 s, and filtered through a filter. Then, mixed with 0.02 M TBA solution (1:1, v/v), reacted at 90 °C, and measured at 532 nm. The values were represented by mg malonaldehyde (MDA) /kg dried oysters.
The sample 4–HNE levels were detected by ELISA kit. Briefly, the standard was diluted on the ELISA–coated plate. The samples were homogenized with a given amount of PBS (pH 7.4). Samples were centrifuged at 2500 rpm for 20 min. The 10 μL supernatant and 40 μL sample diluent were added to the well. The sample was diluted and incubated at 37 °C for 30 min. Enzyme labeling reagent was added to each well, then incubated and washed with a washing solution. The chromogenic reagent was added, and the reaction was terminated after 15 min. The absorbance value of each well was measured at 450 nm with a microplate reader.

Round 2
Reviewer 3 Report
The revised manuscript can now be accepted.